# Cost minimization analysis of treatments for metastatic HER2-positive breast cancer in Peru: Fixed-dose combination of pertuzumab and trastuzumab for subcutaneous injections

**Miguel Figallo** [1]*, **María F. Delgado**[2], **Mauricio Gonzalez**[1], **Adrián Arenas**[3]

**1** APOYO Consultoría, Lima, Peru, **2** London School of Economics, London, United Kingdom, **3** Universidad de Piura, Lima, Perú

* mfigallo@apoyoconsultoria.com

**Data Availability Statement:** All data used to assess costs of anti-Her2 treatments has been

## Abstract

The main objective of this study is to determine whether the employment of fixed-dose combination of pertuzumab and trastuzumab for subcutaneous injection (PH FDC SC; and Phesgo as brand name) to treat metastatic HER2-positive breast cancer patients would minimize costs compared to the traditional treatment of separate intravenous doses of pertuzumab and trastuzumab in Peru. To achieve this, we used EsSalud (the social security health insurance) data and assessed it through a mixed strategy, which consisted of a quantitative and a qualitative approach. The first one aimed to calculate the direct (non-drug consumables, drugs, and healthcare professionals) and indirect costs of both treatments to develop a comparison, whilst the second aimed to validate information and internalize the procedure in an EsSalud context. Overall, we found that the usage of PH FDC SC would be cost saving in EsSalud's context. Specifically, we found three main advantages. Firstly, PH FDC SC generates a savings of 62% in non-drug consumables, which helps alleviate the healthcare system budget constraint. Secondly, its adoption frees up 61 hours of treatment and observation time for a single patient per year, which in turn increases the attention capacity of the healthcare system in terms of nursing hours and chemotherapy couches. Thirdly, the reduction of clinic time supposes an advantage for the patient in the form of increased productivity and well-being. Hence, the adoption of this drug would improve the quality of life of patients while reducing costs and pressure on the healthcare system. This is aligned with the strategy of prioritizing the appropriate breast cancer treatment within the National Cancer Care Plan. In this regard, we also found that the savings produced from switching from the traditional intravenous treatment to the subcutaneous one would allow EsSalud to afford full annual costs of 2 additional treatments, but without increasing their budget. This would cover 7% of the gap of 29 patients who do not have access to full treatment.

uploaded into Data Dryad: https://datadryad.org/stash/share/IulkMs_PFFa5eKfR_51hxMNAfcQJfoI7rhrhkv_UWT4. This does not include third party data that we are not allowed to share publicly.

**Funding:** This study was funded by Roche Peru (https://www.roche.com.pe/) through consulting fees provided to APOYO Consultoría, a company where Miguel Figallo and Mauricio Gonzalez are currently employed. Both Maria Fernanda Delgado and Adrián Arenas were also employed by APOYO Consultoría during the study's development, and their participation was likewise supported by this funding. However, this funding did not increase the salary of the researchers. In addition, the funders had no role in study design, data collection and analysis, decision to publish, or preparation of the manuscript. The funding from Roche Peru solely contributed to the financial support necessary for conducting the study and for paying publication fees. No additional external funding was received for this study.

**Competing interests:** The authors have read the journal's policy and have the following competing interests: MF and MG consulted for Roche Peru through their employment with APOYO Consultoría. This does not alter our adherence to PLOS ONE policies on sharing data and materials. There are no patents, products in development or marketed products associated with this research to declare.

## Introduction

In Peru, breast cancer is the second most common neoplasm, affecting mainly women over 34 years of age. By 2022, 3,166 new cases of breast cancer were reported in Peru, 18% more than the previous year [1]. Within the subtypes of breast cancer, HER2-positive is a type of breast cancer whose cells have higher levels of the HER2 protein, which is present on the outside of all breast cells and promotes their growth [2]. According to the Peruvian Social Security (EsSalud), this type represents about 25% of all cases of breast cancer [3] and it is related with poor prognosis and aggressiveness.

The standard treatment applied by EsSalud for this type of neoplasm consists of the application of anti-HER2 medications—pertuzumab and trastuzumab—which are infused separately into a vein (IV). According to the in-depth interviews, a rough approximation of 30% of patients with HER2-positive metastatic breast cancer are not receiving the pertuzumab and trastuzumab treatment regime. This dual treatment scheme for patients with HER2-positive breast cancer in metastatic stage who have not received previous systemic therapy was approved by Essalud in 2017 [3].

The IV administration has major disadvantages, mainly on the patient's side, in terms of the large number of hours they must spend at the healthcare facility, as well as the considerable discomfort its application entails. Similarly, being a long treatment that requires many supplies, it generates significant costs for the healthcare facility, both in the price of consumables and in the time spent by healthcare professionals.

The fixed-dose combination of pertuzumab and trastuzumab for subcutaneous injection (PH FDC SC; and Phesgo as brand name) has come up as an alternative treatment method for metastatic HER2-positive breast cancer. This treatment utilizes the same drugs as in the standard of care—pertuzumab and trastuzumab—, however, they are administered via a subcutaneous injection (SC), which adds an enzyme called hyaluronidase.

The literature surrounding this topic has previously addressed the advantages of employing the pertuzumab and trastuzumab combined treatment, additional to the cost minimization due to transitioning from the IV to the SC treatment.

Regarding the advantages of PH FDC SC, works such as the FeDeriCa study, have shown that this treatment has the same level of efficacy and offers key advantages when compared to the IV treatment (IV) [4]. Additionally, the PHranceSCa study showcases that breast cancer patients prefer the subcutaneous treatment over the IV scheme [5]. That is because it reduces clinic time and enhances comfort during administration.

Hence, based on the literature, the following benefits are generated by the application of PH FDC SC:

- *On the patient's side*, it presents much shorter administration and observation times, which has reduced the impact of undergoing cancer treatment on their daily lives, as it increases the amount of time they can devote to productive activities, leisure time, and family time, among others [6]. Besides, the application of SC treatment is considerably less intrusive and uncomfortable [5].

- *On the healthcare facility side*, it has been possible to release capacity in the chemotherapy units for other treatments, significantly reduce intravenous compounding costs, and reduce wastage. On the other hand, it saves costs for the healthcare provider, using less active employee time and reducing non-drug consumable costs [6].

As it has been shown, PH FDC SC appears to have similar efficacy and safety profile when compared to the IV scheme, making it reasonable to determine whether it minimizes the treatment costs in EsSalud and in which magnitude. Hence, this study aims to identify savings

opportunities for both patients and healthcare facilities, so that the public expenses execution is optimized without deterring the quality of attention. This is particularly important in the context of high limitations and deficiencies present in the Peruvian healthcare system.

## Materials and methods

### Data used

We use public information from the Peruvian social health security. We use the following sources of data:

### Hospital administrative documents

We retrieved the annual salaries of nurses and pharmaceutical chemists and accessed the list of EsSalud strategic purchases [7]. The latter list included the prices and specific characteristics (size, type, etc.) of drugs, non-drug consumables, and medical equipment for cancer treatment. Finally, the pertuzumab and trastuzumab monthly consumption series and the number of chemotherapy couches were provided by EsSalud. The latter pieces of information were at the hospital level and allowed us to estimate both the number of treated patients and the nationwide attention capacity.

### Clinical practice guidelines

The HER2-positive metastatic and non-metastatic breast cancer clinical practice guidelines contain the step-by-step process and medication needed for the IV treatment [8]. This information served as an input for the in-depth interviews that were followingly conducted.

### Methodology

The study followed a mixed strategy, which considered both a quantitative and qualitative approach. Under the quantitative approach, following recent contributions, we considered a time horizon of a full year and a singular patient approach. Given that the treatment is applied every 21 days, we estimate the cost of a total of 18 cycles or doses per year, which is consistent with previous contributions [6, 9, 10]. Accordingly, for that period of treatment, we defined 3 different stages for HER2-positive metastatic breast cancer:

i. **Loading** *(1 cycle)*: It refers to the initial cycle of each treatment, the following cycles are known as maintenance. Chemotherapy is applied in this cycle along with trastuzumab and pertuzumab by IV infusion separately or both together by PH FDC SC injection subcutaneously.

ii. **Maintenance with chemotherapy** *(5 cycles)*: For the first 5 maintenance cycles, chemotherapy must be applied, along with pertuzumab and trastuzumab.

iii. **Maintenance with no chemotherapy** *(12 cycles)*: For the rest of the 12 maintenance cycles, the treatment is with trastuzumab and pertuzumab. No chemotherapy is required.

For the calculation of costs, we considered the direct and indirect costs. For direct costs, the following three categories were included: (i) non-drug consumables, (ii) drugs, and (iii) healthcare professionals. In the case of indirect costs, we considered an estimation of the value of time of patients to get the savings or losses attributable to the change in the treatment scheme. This calculation used as an input the statistic value of life [11].

On the qualitative side, two in-depth interviews with EsSalud personnel from its largest hospital (Edgardo Rebagliati Martins) in terms of patients and consumption of pertuzumab were carried out

The first interview, performed to a nurse specialist in oncology whit more than 20 years of experience who has been been head of the chemotherapy ward of the Edgardo Rebagliati Martins National Hospital, covered the treatment room procedures. The second one, done to a pharmaceutical chemist in charge of the Oncological Mixtures Unit (UMO) of the same hospital, provided a closer approximation to the oncology mixing room procedures, both executed simultaneously. Both interviews validated information such as: (i) the duration of each identified procedure, and (ii) the specific inputs needed for the execution of each procedure. Those pieces of information were provided as ranks, which let us internalize the variability that time and quantities can suffer when administrating the treatment in the EsSalud context.

## Treatment schemes characterization

The most widely used targeted anti-HER2 treatment is intravenous pertuzumab and trastuzumab (IV). In the case of the Peruvian health system, chemotherapy sessions are applied throughout the treatment cycle [12]. Based on local clinical guidelines and on relevant literature, validated through the interviews with the healthcare professionals, an orderly structure of the steps of the IV treatment for HER2-positive metastatic breast cancer applied by EsSalud was constructed. In addition, the structure for SC treatment was recovered based on the literature reviewed. They are both illustrated below in Fig 1.

# Results

## Cost minimization

We obtained all the costs involved in the treatment of breast cancer, both for the healthcare facility and for the patient, to make an exhaustive comparison between the total costs that each treatment scheme entails. In that sense, the costs of a year of treatment in terms of drug and non-drug consumables, lost staff capacity and patient hours are also calculated. This section

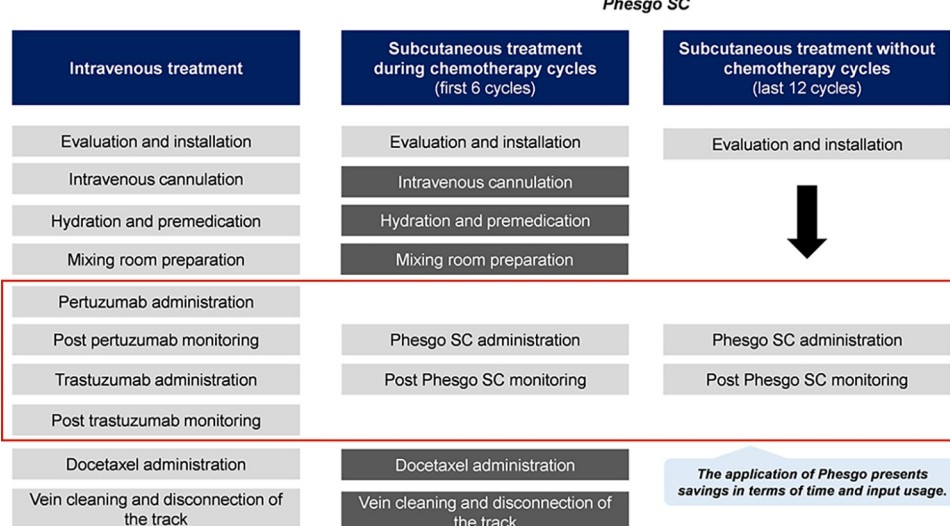

**Fig 1. Stages in IV and SC treatment schemes. Note:** Mixing room preparation procedures occur simultaneously with the previous ones (cannulation, hydration, premedication).

compares the total annual costs of IV and SC treatments and highlights the savings that the SC scheme represents over IV for the following categories: (i) non-drug consumables, (ii) health-care professionals, (iii) indirect costs, and (iv) drug costs.

From previous contributions [6], one can find that the non-drug saving in the United States is of 75%, which is mainly driven by chair time costs (62% of total savings, which equals 75% in this study). For our case, in Fig 2, the total annual costs per patient for each category, excluding drug costs, are shown for both the IV and SC schemes, and the estimated difference between the two is calculated. It is shown that the total annual savings per patient is US$ 5,395, which represents a reduction of approximately 64% of the cost of the IV scheme without including drug costs.

It is worth stressing, however, that this estimation does not represent the overall saving of PH FDC SC, nor does it represent how much can the price of PH FDC SC be increased while still minimizing costs. The correct interpretation to this estimation contemplates that this is the savings in non-drug costs.

Followingly, in Table 1 a breakdown of each category including drug costs is made to analyze in detail the total savings per patient for each of them and which are the main sources—materials or procedures—through which these savings are perceived. A purchase period for nun-drug consumables and drugs is considered between September 5 and 7, 2022.

**Indirect costs.** For the estimation of indirect costs, we considered the patients' loss of productivity generated by the time in clinic (waiting and treatment time). To reach that estimation, we calculated both the time in clinic and the value of patients' time. The following steps were followed:

- **For estimating the time in clinic:** Based on the clinical practice guidelines and the in-depth interviews, we calculated the total time spent in clinic by each patient in each treatment cycle.

- **For the value of patients' time:** We took advantage of the statistic value of life [11] which provides a vector of statistic values for each year of life going from 0 to 99. These calculations

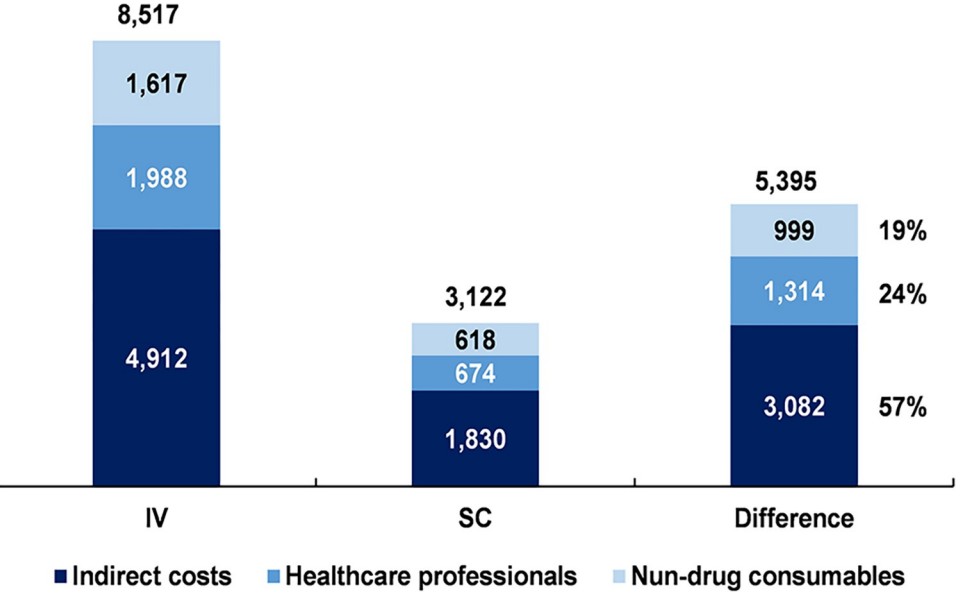

**Fig 2. Annual cost per patient without drug costs, IV vs SC (US$).**

**Table 1. Annual savings per category (US$).**

| | Category | Savings per patient | Main sources of savings |
|---|---|---|---|
| **Direct costs** | | | |
| 1 | Healthcare professionals | US$ 1314 | • Preparation in mixing room (US$ 394) <br>• Administration of monoclonal antibodies (US$ 272) <br>• Hydration (US$ 227) <br>• Monitoring after monoclonal antibodies (US$ 201) |
| 2 | Non-drug consumables (materials) | US$ 999 | • Short double-lumen catheter (US$ 812) <br>• Venoclysis equipment (US$ 91) <br>• Trifurcated equipment (US$ 68) |
| 3 | Drug costs | US$ 332 | • Monoclonal antibodies (US$ 306) <br>• Sodium chloride 250 cc (US$ 15) <br>• Sodium chloride 1 L (US$ 7) |
| **Indirect costs** | | | |
| 4 | Indirect costs | US$ 3,082 | • Administration of monoclonal antibodies (US$ 911) <br>• Hydration (US$ 762) <br>• Monitoring after monoclonal antibodies (US$ 673) <br>• Vein cleaning (US$ 355) |

**Note:** This list is not exhaustive. It only considers the main sources of savings. Hence, the sum of the sources of savings will not necessarily equal the savings per patient shown.

were based on the human capital method, which means that the values of life provided represent the total productivity of a representative individual at each year of age.

By taking advantage of the number of ambulatory attentions related to breast cancer available at the REUNIS database, we obtained weights for estimating an average statistic value of life adjusted to the representative breast cancer patient.

After obtaining this weighted average, we were able to monetize the time saved as a consequence of the use of PH FDC SC, which yielded a total saving per patient of US$3,082. This saving represents a reduction of 63% when compared to the IV scheme.

The main savings are perceived in the stages of: (i) administration of monoclonal antibodies, (ii) hydration, (iii) monitoring after monoclonal antibodies and (iv) vein cleansing, which account for 88% of the total savings in this category. The reasons behind the savings are the same as for the healthcare professional's category; however, in this case, the preparation in mixing room does not come into play since this stage does not require patient time.

**Healthcare professionals.** Regarding the time saved by the healthcare professionals, the time spent in each procedure of both treatments was collected from the interviews and clinical guidelines. Based on those inputs, the nurse-minute costs were calculated (US$ 0.25). As a result, a total saving of $1,314 in healthcare professionals' costs was estimated. This represents a reduction of 66% with respect to the IV scheme.

The main savings come from the stages of preparation in the mixing room, followed by administration of monoclonal antibodies, hydration and monitoring after monoclonal antibodies. These savings collectively represent 83% of the total savings in this category. Since in the SC treatment there are several procedures that are not applied after the first 6 cycles, as can be seen in Fig 1. In addition, other procedures require nurses to dedicate less time to the treatment, such as the administration and monitoring of pertuzumab and trastuzumab.

Table 2 shows the savings in terms of hours of healthcare professionals in the SC scheme compared to the IV scheme.

Although these savings do not directly reduce the budget, they represent a release of human resources capacity for the attention of other patients. Furthermore, this will alleviate the

Table 2. **Hours spent by healthcare professionals for the treatment of a single patient, by stage per year.**

| Stage of treatment | IV | SC | Savings |
|---|---|---|---|
| Loading (1 cycle) | 7.5 | 4.6 | 39% |
| Maintenance with chemotherapy (5 cycles) | 30 | 21.5 | 28% |
| Maintenance with no chemotherapy (12 cycles) | 60 | 10 | 83% |
| **Total hours spent per year** | **97.5** | **36.1** | **63%** |

**Note:** For both treatment schemes, a constant initial evaluation period of 30 minutes per cycle is considered in the total hours spent.

overburden and distress that the healthcare personnel currently go through, which will in turn improve the overall quality of the service. The benefits of this attention capacity release are addressed afterwards.

**Non-drug consumables.** Regarding the non-drug consumables, the main inputs for the analysis were: (i) prices, which were obtained from the EsSalud's catalogs; (ii) quantities required for each treatment scheme, which were collected from clinical guidelines and interviews. As a result of the analysis, a total saving of US$999 in non-drug consumables per patient was estimated. This represents a reduction of 62% with respect to the IV scheme.

The main saving comes from the reduction in the use of: (i) short double-lumen catheter, (ii) venoclysis equipment, and (iii) trifurcated equipment. The savings generated by the disuse of these three elements represent 97% of the total saving in this category, this is because in the SC treatment these instruments are used only in the first 6 cycles when chemotherapy is applied, while in the IV treatment they are used in all cycles. Minor savings are also perceived because in the IV treatment additional consumables are used in the mixing room for the preparation of pertuzumab and trastuzumab, which do not apply to the SC treatment.

A table is presented showing the annual costs in non-drug consumables represented by the inputs of the IV and SC treatment schemes. In that table, considerable savings can be observed in the SC scheme with respect to the IV scheme (Table 3).

As Table 3 shows, the savings are not uniformly distributed across the defined stages. This is so because, in the SC treatment scheme, the catheter is only used in the first 6 cycles when chemotherapy is applied, whilst in the IV scheme, it is used throughout the entire treatment. In addition, since several stages of the SC scheme are only applied in the chemotherapy cycles, as shown in Fig 1, the healthcare facility experiences significant savings in terms of non-drug consumables.

**Drug costs.** The use of PH FDC SC in this treatment also reduces drug costs. This includes the costs and quantity of medicines used in both treatment schemes: pertuzumab, trastuzumab, docetaxel, sodium chloride, and ondansetron, among others. Even under the assumption that PH FDC SC price is equivalent to the sum of prices available of pertuzumab and biosimilar trastuzumab, there is a reduction of US$ 332. This represents a saving of less than 1% on drug costs.

Table 4 showcases that the main savings come from the application of monoclonal antibodies. On one hand, particularly, from the extra trastuzumab and pertuzumab that is required by the IV scheme when reconstituting the drugs. It is worth mentioning that the trastuzumab vial purchased by EsSalud is standardized and contains 21 mg/ml x 20 ml. Since the trastuzumab dose depends on the weight of the patient, more or less than one vial may be required. This parameter was estimated considering a women average weight of 57 kg [13]. Specifically, for the loading cycle, 8mg/kg are applied; for the maintenance cycles, 6 mg/kg are applied. Having a standardized vial of 420 mg and considering the average weight, 456 mg (1.09 vials) would be

**Table 3. Annual costs and savings from non-drug consumables (US$).**

| | Unit cost | First 6 cycles | | | Last 12 cycles | | | Total | Total |
|---|---|---|---|---|---|---|---|---|---|
| | | Costs | | Savings | Costs | | Savings | Savings | Savings |
| **(i) Non-drug consumables** | | IV | SC | IV-SC | IV | SC | IV-SC | IV-SC | % |
| Short double lumen catheter 2 FR | 97 | 406 | 406 | - | 812 | - | 812 | 812 | 67% |
| Venoclysis equipment | 8 | 45 | 45 | - | 91 | - | 91 | 91 | 67% |
| Trifurcated equipment | 6 | 34 | 34 | - | 68 | - | 68 | 68 | 67% |
| PORT catheter needle | 5 | 9 | 9 | - | 17 | - | 17 | 17 | 67% |
| 20 cc syringe | 0 | 2 | 1 | 1 | 3 | - | 3 | 4 | 86% |
| Tegaderm 6x7 | 0 | 2 | 2 | - | 4 | - | 4 | 4 | 67% |
| 18 X 1 1/2" needle | 0 | 1 | 0 | 0 | 1 | - | 1 | 2 | 79% |
| 10 cc syringe | 0 | 1 | 1 | - | 1 | - | 1 | 1 | 67% |
| 5 cc syringe | 0 | 0 | 1 | 0 | 1 | - | 1 | 1 | 50% |
| 20 X 1 1/2" needle | 0 | 0 | 0 | 0 | - | - | - | 0 | (233%) |
| 22 X 1 1/2" needle | 0 | 0 | 0 | - | 0 | - | 0 | 0 | 67% |
| PORT catheter | 287 | 86 | 86 | - | - | - | - | - | - |
| Infusion line | 5 | 32 | 32 | - | - | - | - | - | - |
| Package of gauze | 0 | 0 | 0 | - | 1 | 1 | - | - | - |
| **Total** | **408** | **618** | **617** | **1** | **999** | **1** | **998** | **999** | **100%** |

**Note:** All numbers are rounded to the nearest integer. Prices were retrieved from EsSalud strategic purchases catalog. We use the average exchange rate for the month of December 2022 reported by the Central Reserve Bank of Peru. (PEN/US$: 3.829).

necessary for the loading cycle, whilst 342 mg (0.81 vials) would be necessary for maintenance cycles.

On the other hand, PH FDC SC provides a fixed dose, and its price is equal to the sum of the prices of pertuzumab and trastuzumab, except for the loading cycle, where the price of PH FDC SC is multiplied by a factor of 1.5 (this is consistent with its current global pricing strategy). Hence, spending on a fixed dose of SC results cheaper than spending on the alternative, which may require additional expenditures on trastuzumab depending on the patient's weight.

**Table 4. Annual costs and savings from drugs (US$).**

| | Unit cost | First 6 cycles | | | Last 12 cycles | | | Total | Total |
|---|---|---|---|---|---|---|---|---|---|
| | | Costs | | Savings | Costs | | Savings | Savings | Savings |
| **(i) Drug consumables** | | IV | SC[1/] | IV-SC | IV | SC[1/] | IV-SC | IV-SC | % |
| Dual anti Her2 drugs | 2,128 | 14,552 | 13,830 | 722 | 25,117 | 25,533 | -416 | 306 | 0% |
| Sodium chloride 250cc | 0 | 8 | 3 | 5 | 10 | - | 10 | 15 | 83% |
| Sodium chloride 1L | 1 | 4 | 4 | - | 7 | - | 7 | 7 | 64% |
| Ondansetrón 8 mg | 0 | 2 | 2 | - | 3 | - | 3 | 3 | 60% |
| Dexamethasone 2 mg / ml X 2 ml | 0 | 0 | 0 | - | 1 | - | 1 | 1 | 100% |
| Docetaxel | 16 | 98 | 98 | - | - | - | - | - | - |
| Clorphenamine 10 mg / ml X 1 ml | 0 | 1 | 1 | - | - | - | - | - | - |
| Ranitidine 25 mg / ml X 2 ml | 0 | 0 | 0 | - | - | - | - | - | - |
| **Total** | **2,145** | **14,665** | **13,938** | **727** | **25,138** | **25,533** | **-395** | **332** | **100%** |

**Note:** All numbers are rounded to the nearest integer. Prices were retrieved from EsSalud strategic purchases catalog. We use the average exchange rate for the month of December 2022 reported by the Central Reserve Bank of Peru. (PEN/US$: 3.829). 1/ The SC treatment is referred to PH FDC SC, while the IV treatment refers to the intravenous administration of pertuzumab and trastuzumab.

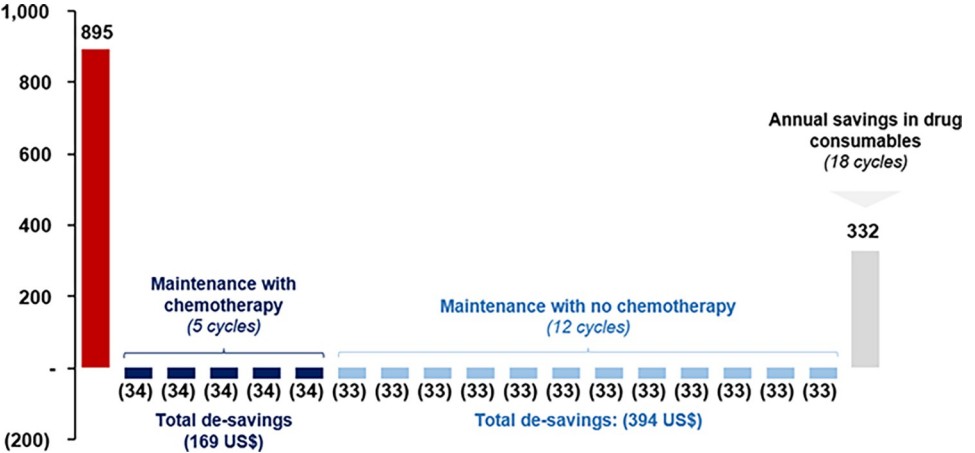

**Fig 3. Savings and de-savings for all 18 cycles in drug costs of using PH FDC SC (US$).**

As it is not plausible to expect a uniform distribution for patients' weight, there is a variable leftover of trastuzumab in the IV scheme. Our model considers an ideal case in which this leftover is never wasted. However, this assumption stresses the results because it reduces the drug costs of the IV scheme, which underestimates the savings generated by the adoption of PH FDC SC. For every 1% of leftover wasted, savings increase in US$ 26, which represents 8% of the estimated drug costs savings. Under the extreme case in which the whole leftover is wasted, savings in drug costs would equal US$ 589.

However, the above-mentioned savings are concentrated in the first 6 cycles of treatment, and, particularly in the loading cycle. Considering an average-weight patient, the first cycle of the IV scheme requires the use of two vials of pertuzumab, which would cost US$ 3,882, and 456 mg of trastuzumab which would cost US$ 203. Thus, the application of monoclonal antibodies under the IV scheme requires an expenditure of US$ 4,085 only for the loading cycle. In comparison, the SC scheme takes advantage of a fixed loading dose of PH FDC SC, which costs US$ 3,192, which implies a saving of US$ 893 only for the first cycle.

The following cycles (2–18) showcase minor de-savings, which are visible in the first row of Table 4. The reason behind those losses is the assumption of no waste of leftovers. As it is detailed in footnote 10, the use of this assumption underestimates the pharmacological savings of PH FDC SC because it considers an ideal setting in which the content of each vial of trastuzumab is perfectly distributed and consumed. Notwithstanding the above, the pharmacological savings resist this extreme scenario, which strengthens the savings capacity of PH FDC SC. Fig 3 shows a breakdown of all savings and de-savings received in all 18 cycles in one year divided by stage.

## Payer's costs

Two steps were followed to estimate the budgetary impact for the payer: (i) estimation of the number of patients treated with complete and incomplete treatment and (ii) monetary and non-monetary savings estimation, which we proceed to detail below.

**Estimated number of patients.** The number of patients was estimated through two different approaches. The first approach (*top-down*) consisted of taking the total cases presented by GLOBOCAN 2020 of breast cancer in Peru (22,486) and disaggregating them progressively by multiplying it by the following probabilities: (i) suffering HER2-positive breast cancer (25–30%), (ii) suffering metastatic breast cancer (9%) reported by FISSAL, and (iii) being treated

by EsSalud (18%, estimated by considering the total breast cancer patients attended by Minsa, EsSalud, and INEN). Fig 4 shows each step and intermediate result of this estimation. As it is shown, we concluded that EsSalud currently treats 100 HER2-positive patients in the metastatic stage.

Additionally, based on the in-depth interviews with key EsSalud personnel, an approximate of 30% of HER2-positive metastatic breast cancer are not treated with pertuzumab, despite requiring it. By considering this proportion of patients who do not have access to full treatment, we calculate that there is a gap of 30 patients who are not receiving proper treatment.

The second approach (*bottom-up*) consisted of obtaining the expected monthly consumption of pertuzumab for a single patient. For that to happen, we used the probability of being a new patient (31%) by obtaining the 5-year prevalence of breast cancer (22,486) and the number of new cases per year (6,860). This probability allowed us to calculate an expected monthly consumption of 1.31 vials, considering that the patient receives each dose once every month (although medical indication states that the dose must be applied every 21 days, we considered the considerable lags in administration). Therefore, this calculation was the result of the weighted sum of 1 vial consumed by a maintenance patient and 2 vials consumed by a new patient. Rescaling this value to an annual time frame yields an expected consumption of 15.66 per patient. Then, the total pertuzumab consumption of each healthcare facility was divided by the expected consumption per patient to get an estimate of the number of patients, which we calculate at 65.

Nonetheless, as this calculation took the consumption of pertuzumab as an input, we rescaled it to get the full universe of HER2-positive metastatic breast cancer. Specifically, we followed the fact that there is a gap of 30% who do not receive full treatment, which means that this 30% only receive trastuzumab.

This exercise allowed us to estimate that there was a full universe of 93 patients treated with either complete or incomplete treatment schemes, which yields a gap of 28 patients that do not have access to full treatment. Fig 5 shows each step and intermediate result of the described approach.

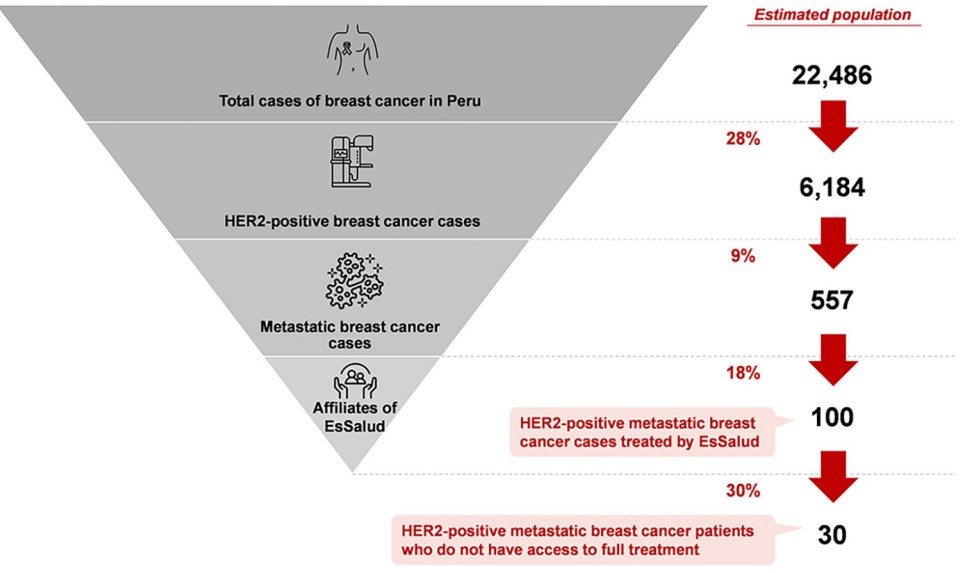

**Fig 4. Top-down approach.**

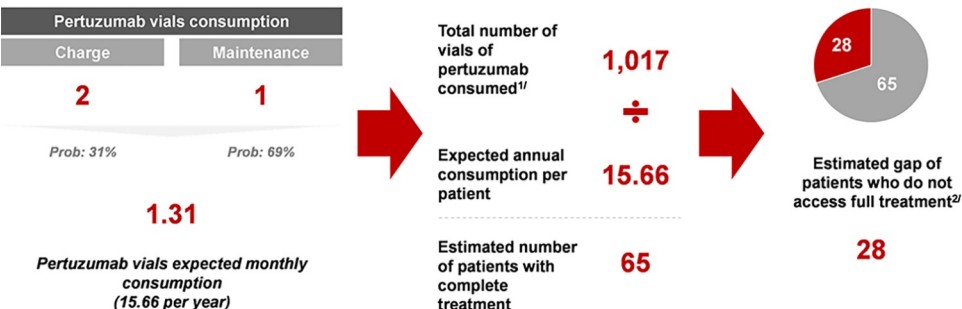

**Fig 5. Bottom-up approach. Notes:** 1/ Total number of vials of pertuzumab consumed in EsSalud during 2022. Recovered from the list of purchases of strategic goods. 2/ The estimation of the gap follows the fact that the number of patients with complete treatment are the 70% of the full universe of attended patients. Source: In-depth interviews.

Based on the above-described methodologies, it is estimated that the average gap of patients who do not access to full treatment is of 29, which, according to the information recovered in the interviews, can be explained by the restrictions and resource limitations that challenge the healthcare system.

**Monetary and non-monetary savings for EsSalud.** Based on the above estimates, the total EsSalud monetary savings in direct costs at the national level are US$ 90,535. This estimation considers both approaches' average estimate of patients who currently access full treatment in EsSalud (68), it is calculated that the social security would save US$ 67,947 in non-drug consumables per year by adopting PH FDC SC. This represents 62% of what the social security system spends each year on consumables needed for the treatment of HER2-positive metastatic breast cancer. Additionally, the use of PH FDC SC generates savings in drug costs, as the SC treatment demands lower uses of premedication, hydration, and reconstitution. These savings are of US$ 22,588 per year. Moreover, a sensitivity analysis was performed, which showcased that an increase of 0.84% in the price of PH FDC SC is required for the savings in drug costs to be zero, and an increase of 3.38% is required for the savings in direct costs —which includes drug costs and non-drug consumables costs—to be zero.

Moreover, the adoption of the subcutaneous treatment would also have non-monetary savings in terms of (i) the availability of nurses and (ii) the release of chemotherapy couches use time. Since the adoption of PH FDC SC generates an annual saving of 61 hours per patient and we estimate 68 currently attended patients, it is calculated that there will be a release of more than 4,000 nursing hours and chemotherapy chairs use time. As we estimate that 97 hours are required to attend a patient through a whole year under an IV scheme, we calculate that the freeing up of capacity would allow the attention of 41 additional IV patients per year without incurring on more investment costs than the needed to be able to close the access gap of 29 HER2-positive metastatic breast cancer patients.

## Robustness checks

To stress our results, several robustness checks were performed. In particular, we ran sensitivity analyses for two key variables: (i) price of PH FDC SC and (ii) price of Pertuzumab. Regarding the first test, Fig 6 provides three different scenarios that allow us to tell how total savings for a single patient are affected by the price of the SC treatment. Those scenarios are: (i) baseline, (ii) 5% increase in the price of PH FDC SC, and (iii) 5% decrease in the price of PH FDC SC without adjusting the price of pertuzumab and trastuzumab.

A 5% increase in the price of PH FDC SC would reduce total savings in 34%, going from US$ 5,727 to US$ 3,759 per patient. This result features a de-savings of US$ 1,636 per patient

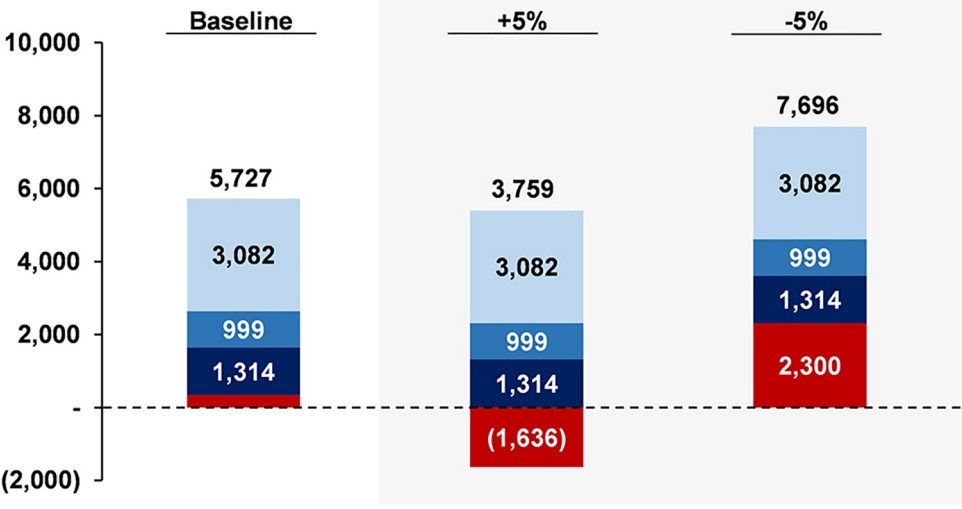

**Fig 6. Price of PH FDC SC sensitivity analysis for the annual savings of a single patient (US$). Note:** For the construction of each scenario, every other variable is kept constant. The savings here portrait consider both the experimented by patients and the payer in a timeframe of a full year.

in drug costs, which runs up to US$ 111,246 when rescaling it to the full universe of attended patients. However, what is more important is that, despite bearing this loss, total savings are positive. That is because we are not only considering direct savings for the payer–which are, in this case, negative–, but savings in healthcare professionals and patients time, whose monetary equivalents are featured in Fig 6.

Besides, we find an expansion of savings when considering a 5% decrease in the price of PH FDC SC. Annual savings in drug costs go up to US$ 2,300 per patient, which is equivalent to a total saving of US$ 156,421 when rescaling it to the full social security system. This would have two strong benefits: (i) from a social perspective, savings experiment an increase of 34%, and (ii) from a payer's perspective, savings experiment an increase of 148%.

As to the sensitivity analysis with respect to the price of pertuzumab, Fig 7 shows three different tests that were conducted. Each of these tests consisted in the progressive addition of

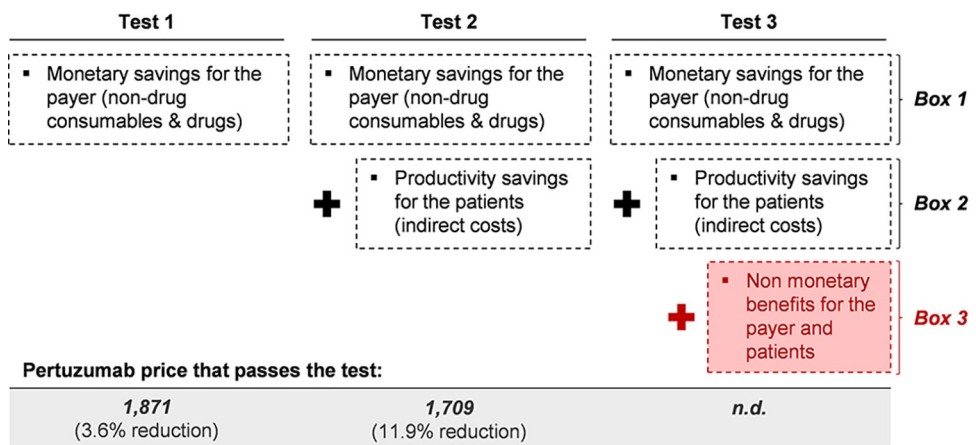

**Fig 7. Pertuzumab price sensitivity analysis. Note:** The baseline price for pertuzumab is US$ 1,941. There is no estimation for test 3, as it contemplates a box of non-monetary savings.

savings boxes with the aim of finding the reduction in the price of pertuzumab that offsets the savings derived from the application of PH FDC SC. In particular, the first test considers up to box 1, which includes only the monetary savings for the payer. That is, the savings in drugs and non-drug consumables, which are equal to US$ 90,534. We find that a 3.6% decrease in the price of pertuzumab is sufficient to equal these savings.

The second test considers up to box 2, which adds patients' savings in indirect costs. It is worth it to recall that these savings only considered measurable benefits that were monetizable, which are savings in productivity. These savings were calculated in US$ 209,556. In that sense, the second test consisted in finding a price of pertuzumab that let us offset the US$ 300,090 savings that are generated by boxes 1 and 2. We find that, for that to happen, there must be a 11.9% decrease in the price of pertuzumab.

Finally, we add a referential test–test 3– to stress the fact that there is a third box of savings that no reduction in the price of pertuzumab could offset. This group of savings contemplates non-monetary benefits both for patients and the payer. Regarding the patients, there are important benefits such as: gain in comfort, gain in time for no productive activities, gain in overall utility. On the other hand, for the payer, there are other such benefits like: attention capacity increase, relief of overburden for nurses, among others.

## Conclusions and discussion

As it was shown, switching from an IV to a SC treatment scheme produces significant monetary and non-monetary savings for the healthcare system and the patient. PH FDC SC generates a savings of 62% in non-drug consumables, which helps alleviate the healthcare system budget constraint. Likewise, its adoption results in a reduction of 61 hours in treatment and observation time for a single patient per year. This supposes an advantage for both the payer and the patient. For the payer, it frees up nursing hours and chemotherapy couch availability; for the patient, it reduces time in the clinic, resulting in increased productivity and well-being. This time-saving benefits that PH FDC SC provides can be extrapolated to early-stage treatment scenarios such as neoadjuvant and adjuvant, as our methodology considers 6 initial cycles where the treatment is complemented by chemotherapy.

Despite this result, we estimate that the generated monetary savings allow EsSalud to afford the full annual costs of only 2 additional treatments without increasing its budget. This would cover 7% of the gap of 29 patients who do not have access to full treatment without increasing the budget.

Furthermore, increasing the number of treated patients is only possible through the adoption of PH FDC SC instead of the current IV treatment. This is mainly because the use of PH FDC SC decreases the amount of infrastructure and personnel required to treat each patient. Otherwise, EsSalud would be required to invest in expanding the existing infrastructure to be able of closing the treatment gap, due to the constraints of healthcare resources. Therefore, even if biosimilars were to match the monetary savings of PH FDC SC (drug and consumables costs) and monetary savings were achieved, there would be no capacity to expand the current care.

Hence, in terms of health strategies, these results suggest that if EsSalud were to adopt the use of PH FDC SC, expenditures and pressure on the health system would be mitigated. Moreover, this relief for the system would be accompanied by an improvement in the Her2-positive metastatic breast cancer patients, who would access a less intrusive treatment that demands less administration and observation times than the current standard of care. This is aligned with the strategy of prioritizing breast cancer within the National Integral Cancer Care Plan [14].

In a similar study conducted in Cuba, it was found that the subcutaneous treatment proved to minimize costs with respect to the intravenous treatment even when its price was increased by 30% [9]. This result differs from what we find (15.83%) because the characteristics of both healthcare systems have critical differences.

On the one hand, the prices of the IV and SC anti-Her2 drugs are considerably higher than what we are considering. In fact, almost all the savings that study found were explained by that source. On the other hand, the non-drug consumables that are considered in the Cuban study are considerably cheaper than what is considered in this study, which discards non-drug consumables as a relevant source of savings. Finally, the Cuban estimation for IV procedures duration is higher than ours, which explains why their time savings are higher than ours.

Some factors that limit the scope of the results include the number of conducted interviews. In response to this, an interesting extension would be the incorporation of parameters associated with other hospitals in the EsSalud network with the aim of verifying the presence of heterogeneity in the cost minimization results. Similarly, this study does not delve into the availability of settings for SC treatment administration. Although we conclude that the adoption of PH FDC SC will free up chemotherapy couches, it would be interesting to validate that these settings can be arranged for SC treatment.

Finally, it is important to indicate that the price of PH FDC SC considered in this study is referential. At the date of the analysis, it was not considered whether it is subject to taxation or not. However, we consider that this matter does not interfere with the conclusions of this study, since taxes return to the public treasury in the long run, so it does not represent an additional cost for the payer.

Additional extensions to this study may cover a deeper qualitative approach, such as the collection of patients' perceptions on which treatment increase their welfare. In the same way, we recommend future contributions to carry out the stopwatch measurement of the mixing and treatment room procedures, and the inclusion of further welfare components in the indirect costs, which we delimit to the loss of productivity. Finally, it would be relevant to test if the above conclusions are extensible to other Peruvian healthcare providers, such as the ones administered by the Ministry of health (Minsa).

## Acknowledgments

We thank Rodrigo Castro for his help format this research paper for publication. His contribution ensured the professional presentation of this work.

## Author Contributions

**Conceptualization:** Miguel Figallo, Mauricio Gonzalez.

**Data curation:** Miguel Figallo, María F. Delgado, Mauricio Gonzalez, Adrián Arenas.

**Formal analysis:** María F. Delgado, Mauricio Gonzalez, Adrián Arenas.

**Funding acquisition:** Miguel Figallo.

**Investigation:** Miguel Figallo, María F. Delgado.

**Methodology:** Miguel Figallo, María F. Delgado.

**Project administration:** Miguel Figallo.

**Supervision:** Miguel Figallo.

**Validation:** Miguel Figallo, Mauricio Gonzalez.

**Visualization:** Miguel Figallo, Mauricio Gonzalez.

**Writing – original draft:** Miguel Figallo, María F. Delgado, Mauricio Gonzalez.

**Writing – review & editing:** Miguel Figallo, Mauricio Gonzalez, Adrián Arenas.

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
