## [Decision Letter · Decision Letter 0]

22 May 2024

PONE-D-23-39363Cost minimization analysis of treatments for metastatic HER2-positive breast cancer: Fixed-Dose combination of pertuzumab and trastuzumab for subcutaneous injectionsPLOS ONE

Dear Dr. Figallo,

Thank you for submitting your manuscript to PLOS ONE. After careful consideration, we feel that it has merit but does not fully meet PLOS ONE’s publication criteria as it currently stands. Therefore, we invite you to submit a revised version of the manuscript that addresses the points raised during the review process.

**ACADEMIC EDITOR:** Please consider carefully the following aspects: 

Introduction

HER2-positivity is associated with more aggressive disease when anti-HER2 treatment is not given. I suggest the authors to revise their sentence about that.What in-depth interviews are behind the sentence of 30% of patients not receiving pertuzumab? This part needs clarification.When the authors describe the advantages of Phesgo, relevant studies related to clinical efficacy and safety should be cited.In general, the introduction is quite long with some parts to be more suitable for Discussion. I suggest the authors to make the introduction more concise and solid.

Methods

The process for choosing EsSalud personnel that was included in the qualitative part of the study should be described.The number of EsSalud personnel should also be included.

Results

The manuscript included a lot of tables and figures. Many of the tables can be included as Supplementary material (as Table 4, Table 7, figures 4, 5 and 7) or deleted (table 6)

Discussion

In the first paragraph, the authors included information about their sensitivity analysis on the price of Phesgo and how much it could be increased to cover the savings. This result is not as relevant as the other results to be included in this paragraph.A comparison with similar analyses in similar context as in Peru would be beneficial. Some parts of Introduction could be used here.A part in the discussion dedicated to limitations should be included.An implication of study results to healthcare strategies in Peru would be of interest. This aspect is lacking from the abstract as well.

General comment

The figures should be uploaded in better resolution.==============================

We look forward to receiving your revised manuscript.

Kind regards,

Antonis Valachis

Academic Editor

PLOS ONE

Journal Requirements:

 [This study was funded by Roche Peru. However, it is important to note that the study was conducted independently, and Roche Peru did not participate in the design, execution, or analysis of the research. The funding from Roche Peru solely contributed to the financial support necessary for conducting the study. Therefore, the independence of the study was rigorously maintained, and the researchers retained full control over the study design, data collection, analysis, interpretation, and manuscript preparation. Thus, we declare that there are no conflicts of interest related to the funding source, and the results presented in this manuscript are solely based on the scientific merits of the research].  

4. Thank you for uploading your study's underlying data set. Unfortunately, the repository you have noted in your Data Availability statement does not qualify as an acceptable data repository according to PLOS's standards.

Reviewers' comments:

Reviewer's Responses to Questions

**Comments to the Author**

1. Is the manuscript technically sound, and do the data support the conclusions?

Reviewer #1: Yes

2. Has the statistical analysis been performed appropriately and rigorously? 

Reviewer #1: Yes

3. Have the authors made all data underlying the findings in their manuscript fully available?

Reviewer #1: Yes

4. Is the manuscript presented in an intelligible fashion and written in standard English?

Reviewer #1: Yes

5. Review Comments to the Author

Reviewer #1: Dear Author

interesting results explaining the direct and indirect costs with sc administration

Was the study approved by an ethic commitee?

I suggest to substitute the brand name with the name of the substance

6. PLOS authors have the option to publish the peer review history of their article (what does this mean?). If published, this will include your full peer review and any attached files.

Reviewer #1: No

---

## [Author Response · Author response to Decision Letter 0]

22 Jul 2024

On behalf of the team of authors of the study, we are grateful for the comments and queries that have been sent to us. Below, we provide further details on how we have approached each of these:

- On the adjustments to the introduction: References were placed in the texts related to the clinical efficacy and safety of Phesgo, as well as its advantages. The introduction was also summarized.

- On the qualitative data collection strategy: It was detailed in the text (Methodology section) that two interviews were conducted with key personnel from the largest and most equipped hospital in the EsSalud network (Rebagliati). In this section, the credentials and experience of the interviewees were described.

- About tables and figures: Some tables were removed, aligned with what was suggested. Regarding the figures, these were included as supplementary material. In addition, the resolution of the figures was improved.

- On the discussion and conclusions: The Phesgo price sensitivity analysis was removed. In addition, a comparison of the results with a Cuban study applied in a similar context was added. We also expanded our description of the limitations of our study and described the implications of our results for the Peruvian healthcare strategies.

- General comments: The study was not submitted to an ethics committee, as it did not involve a clinical study. In addition, as requested, the trade name of Phesgo was changed to the name of its substance (PH FDC SC).

---

## [Editor Report · Decision Letter 1]

24 Jul 2024

Cost minimization analysis of treatments for metastatic HER2-positive breast cancer in Peru: Fixed-Dose combination of pertuzumab and trastuzumab for subcutaneous injections

PONE-D-23-39363R1

Dear Dr. Figallo,

We’re pleased to inform you that your manuscript has been judged scientifically suitable for publication and will be formally accepted for publication once it meets all outstanding technical requirements.

Kind regards,

Antonis Valachis

Academic Editor

PLOS ONE
---

## [Editor Report · Acceptance letter]

13 Aug 2024

PONE-D-23-39363R1 

PLOS ONE

Dear Dr. Figallo, 

I'm pleased to inform you that your manuscript has been deemed suitable for publication in PLOS ONE. Congratulations! Your manuscript is now being handed over to our production team.

Kind regards, 

on behalf of

Assoc Prof Antonis Valachis 

Academic Editor

PLOS ONE